# The tRNA thiolation-mediated translational control is essential for plant immunity

Xueao Zheng[1,2,3,4,5,6†], Hanchen Chen[1,3,4,5,6†], Zhiping Deng[7], Yujing Wu[1,3,4,5,6], Linlin Zhong[8], Chong Wu[1,3,4,5,6], Xiaodan Yu[1,3,4,5,6], Qiansi Chen[2], Shunping Yan[1,3,4,5,6]*

[1]Hubei Hongshan Laboratory, Wuhan, China; [2]Zhengzhou Tobacco Research Institute of CNTC, Zhengzhou, China; [3]College of Life Science and Technology, Huazhong Agricultural University, Wuhan, China; [4]Shenzhen Branch, Guangdong Laboratory for Lingnan Modern Agriculture, Shenzhen, China; [5]Agricultural Genomics Institute at Shenzhen, Chinese Academy of Agricultural Sciences, Shenzhen, China; [6]Shenzhen Institute of Nutrition and Health, Huazhong Agricultural University, Shenzhen, China; [7]State Key Laboratory for Managing Biotic and Chemical Threats to the Quality and Safety of Agro-products, Institute of Virology and Biotechnology, Zhejiang Academy of Agricultural Sciences, Hangzhou, China; [8]Key Laboratory of Horticultural Plant Biology, Ministry of Education, College of Horticulture and Forestry Sciences, Huazhong Agricultural University, Wuhan, China

**\*For correspondence:**
spyan@mail.hzau.edu.cn

[†]These authors contributed equally to this work

**Competing interest:** The authors declare that no competing interests exist.

**Abstract** Plants have evolved sophisticated mechanisms to regulate gene expression to activate immune responses against pathogen infections. However, how the translation system contributes to plant immunity is largely unknown. The evolutionarily conserved thiolation modification of transfer RNA (tRNA) ensures efficient decoding during translation. Here, we show that tRNA thiolation is required for plant immunity in *Arabidopsis*. We identify a *cgb* mutant that is hyper-susceptible to the pathogen *Pseudomonas syringae*. *CGB* encodes ROL5, a homolog of yeast NCS6 required for tRNA thiolation. ROL5 physically interacts with CTU2, a homolog of yeast NCS2. Mutations in either *ROL5* or *CTU2* result in loss of tRNA thiolation. Further analyses reveal that both transcriptome and proteome reprogramming during immune responses are compromised in *cgb*. Notably, the translation of salicylic acid receptor NPR1 is reduced in *cgb*, resulting in compromised salicylic acid signaling. Our study not only reveals a regulatory mechanism for plant immunity but also uncovers an additional biological function of tRNA thiolation.

## Editor's evaluation

This valuable study provides solid evidence for a role of tRNA thiolation in Arabidopsis immunity through genetic, transcriptomic, and proteomic approaches, specifically that the tRNA mcm5s2U modification affects SA signaling through NPR1 translation.

## Introduction

As sessile organisms, plants are frequently infected by different pathogens, which greatly affect plant growth and development, and cause a tremendous loss in agriculture (*Jones and Dangl, 2006*; *Spoel and Dong, 2012*; *Yan et al., 2013*). To defend against pathogens, plants have evolved sophisticated

immune mechanisms. One essential immune regulator is the phytohormone salicylic acid (SA), which plays a central role in immune responses (*Vlot et al., 2009*; *Peng et al., 2021*; *Yan and Dong, 2014*; *Zhou and Zhang, 2020*). Upon pathogen infection, the biosynthesis of SA is dramatically induced. Plants defective in SA biosynthesis or SA signaling are hyper-susceptible to pathogens (*Cao et al., 1997*; *Rekhter et al., 2019*). Several independent forward genetic screens revealed that NONEXPRESSER OF PR GENES 1 (NPR1) is a master regulator of SA signaling (*Canet et al., 2010*; *Cao et al., 1997*; *Ryals et al., 1997*; *Shah et al., 1997*). In the *Arabidopsis npr1* mutant, the SA-mediated immune responses are dramatically reduced. Biochemical and structural studies suggested that NPR1 and its homologs NPR3 and NPR4 are SA receptors (*Ding et al., 2018*; *Fu et al., 2012*; *Kumar et al., 2022*; *Wang et al., 2020*; *Wu et al., 2012*; *Zhou et al., 2023*).

Immune responses involve massive changes in gene expression at transcription, post-transcription, translation, and post-translation levels. Compared with other regulatory mechanisms, the translation regulation mechanism is less well studied. Notably, it is reported that both the pattern-triggered immunity (PTI) and effector-triggered immunity (ETI) involve translational reprogramming (*Xu et al., 2017*; *Yoo et al., 2020*). And PABP/purine-rich motif was described as an initiation module for PTI-associated translation (*Wang et al., 2022*) and CDC123, an ATP-grasp protein, is a key activator of ETI-associated translation (*Chen et al., 2023b*).

During translation, the code information of mRNA is decoded by transfer RNA (tRNA) molecules, which carry different amino acids. In this sense, the tRNA molecules function as deliverers of the building blocks for translation. The decoding efficiency of tRNAs is affected by their abundance and modifications as well as aminoacyl-tRNA synthetases, amino acid abundance, and elongation factors. Interestingly, an emerging regulatory role for tRNA modifications during elongation has been reported (*Delaunay et al., 2016*; *Schaffrath and Leidel, 2017*; *Torres et al., 2014*).

Currently, more than 150 different tRNA modifications have been identified (*Agris et al., 2018*). Among them, the 5-methoxycarbonylmethyl-2-thiouridine of uridine at wobble nucleotide ($mcm^5s^2U$) is highly conserved in all eukaryotes. The $mcm^5s^2U$ modification is present in the wobble position of tRNA-Lys(UUU), tRNA-Gln(UUG), and tRNA-Glu(UUC) (*Huang et al., 2005*; *Lu et al., 2005*; *Sen and Ghosh, 1976*). In budding yeast (*Saccharomyces cerevisiae*), the 5-methoxycarbonylmethyl of uridine ($mcm^5U$) is catalyzed by the Elongator protein (ELP) complex and the Trm9/112 complex, whereas thiolation ($s^2U$) is mediated by the ubiquitin-related modifier 1 (URM1) pathway involving URM1, UBA4, NCS2, and NCS6 (*Leidel et al., 2009*; *Nakai et al., 2004*; *Noma et al., 2009*; *Zabel et al., 2008*). Loss of the $mcm^5s^2U$ modification causes ribosome pausing at AAA and CAA codons, which results in defective co-translational folding of nascent peptides and protein aggregation, thereby disrupting proteome homeostasis (*Nedialkova and Leidel, 2015*; *Ranjan and Rodnina, 2017*; *Rezgui et al., 2013*). In yeasts, the $mcm^5s^2U$ modification was reported to regulate cell cycle, DNA damage repair, and abiotic stress responses (*Dewez et al., 2008*; *Jablonowski et al., 2006*; *Klassen et al., 2017*; *Leidel et al., 2009*; *Nedialkova and Leidel, 2015*; *Zinshteyn and Gilbert, 2013*). In humans, loss of the $mcm^5s^2U$ modification causes numerous disorders including severe developmental defects, neurological diseases, tumorigenesis, and cancer metastasis (*Pan, 2018*; *Shaheen et al., 2019*; *Simpson et al., 2009*; *Torres et al., 2014*; *Waszak et al., 2020*). In plants, loss of the $mcm^5s^2U$ modification was associated with developmental defects and hypersensitivity to heat stress (*Leiber et al., 2010*; *Nakai et al., 2019*; *Xu et al., 2020*). However, it remains unknown whether the $mcm^5s^2U$ modification is involved in plant immune responses.

In this study, we found that the $mcm^5s^2U$ modification is required for plant immunity. Transcriptome and proteome analyses revealed that the $mcm^5s^2U$ modification is essential for the reprogramming of immune-related genes. Especially, the translation of the master immune regulator NPR1 is compromised in the $mcm^5s^2U$ mutant. Our study not only expands the biological function of tRNA thiolation but also highlights the importance of translation control in plant immunity.

## Results

### ROL5 is required for plant immunity

In a study to test the disease phenotypes of some transgenic *Arabidopsis*, we found that one transgenic line was hyper-susceptible to the bacterial pathogen *Pseudomonas syringae* pv. *Maculicola* (*Psm*) ES4326. The disease symptom resembled that of *npr1*, in which the master immune

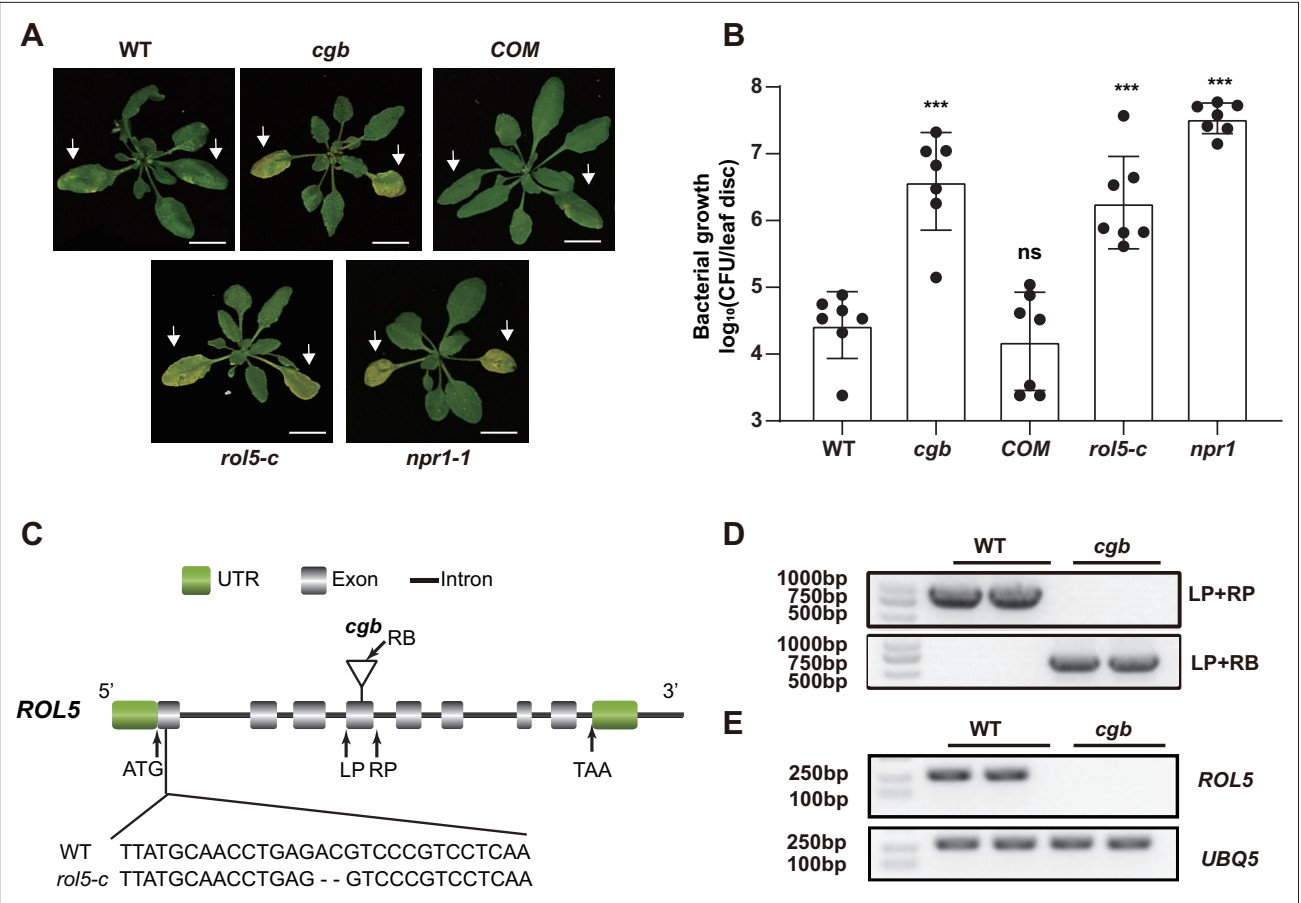

**Figure 1.** The *rol5* mutants are more susceptible to the bacterial pathogen *Psm* ES4326 than wild-type (WT). (**A**) Pictures of *Arabidopsis* 3 days after infection. The arrows indicate the leaves inoculated with *Psm* ES4326 (OD$_{600}$=0.0002). *cgb* and *rol5-c* are mutants defective in *ROL5. COM*, the complementation line of *cgb. npr1-1* serves as a positive control. Bar = 1 cm. (**B**) The growth of *Psm* ES4326. CFU, colony-forming unit. Error bars represent 95% confidence intervals (n=7). Statistical significance was determined by two-tailed Student's t-test. ***, p<0.001; ns, not significant. (**C**) A schematic diagram showing the site of the T-DNA insertion in *cgb* and the deleted nucleotides in *rol5-c*. (**D**) The genotyping results using the primers indicated in C. (**E**) The transcript of *ROL5* is not detectable in *cgb. UBQ5* serves as an internal reference gene.

The online version of this article includes the following source data for figure 1:

**Source data 1.** Source data related to *Figure 1B*.

**Source data 2.** Source data related to *Figure 1D*.

**Source data 3.** Source data related to *Figure 1E*.

regulator NPR1 was mutated (*Figure 1A and B*). We named this line *cgb* (for Chao Gan Bing; 'hyper-susceptible to pathogens' in Chinese). To identify the causal gene of *cgb*, we sequenced its genome using the next-generation sequencing technology, which revealed that there was a T-DNA insertion in the fourth exon of *ROL5* (AT2G44270; *Figure 1C*). The insertion was confirmed by genotyping analysis (*Figure 1D*). In the *cgb* mutant, the transcript of *ROL5* was undetectable (*Figure 1E*), indicating that *cgb* was a knock-out mutant. To confirm that *ROL5* was the *CGB* gene, we carried out a complementation experiment by transforming *ROL5* into the *cgb* mutant. As shown in *Figure 1A and B*, the disease phenotype of the complementation line (*COM*) was similar to that of wild-type (WT). Moreover, we generated another allele of *ROL5* mutant, *rol5-c*, using the CRISPR-Cas9 gene-editing approach (*Wang et al., 2015*). In *rol5-c*, a 2 bp deletion in the first exon of *ROL5* causes a frameshift (*Figure 1C*). As expected, the *rol5-c* mutant was hyper-susceptible to *Psm* as *cgb* (*Figure 1A and B*). These data strongly suggested that ROL5 is required for plant immunity.

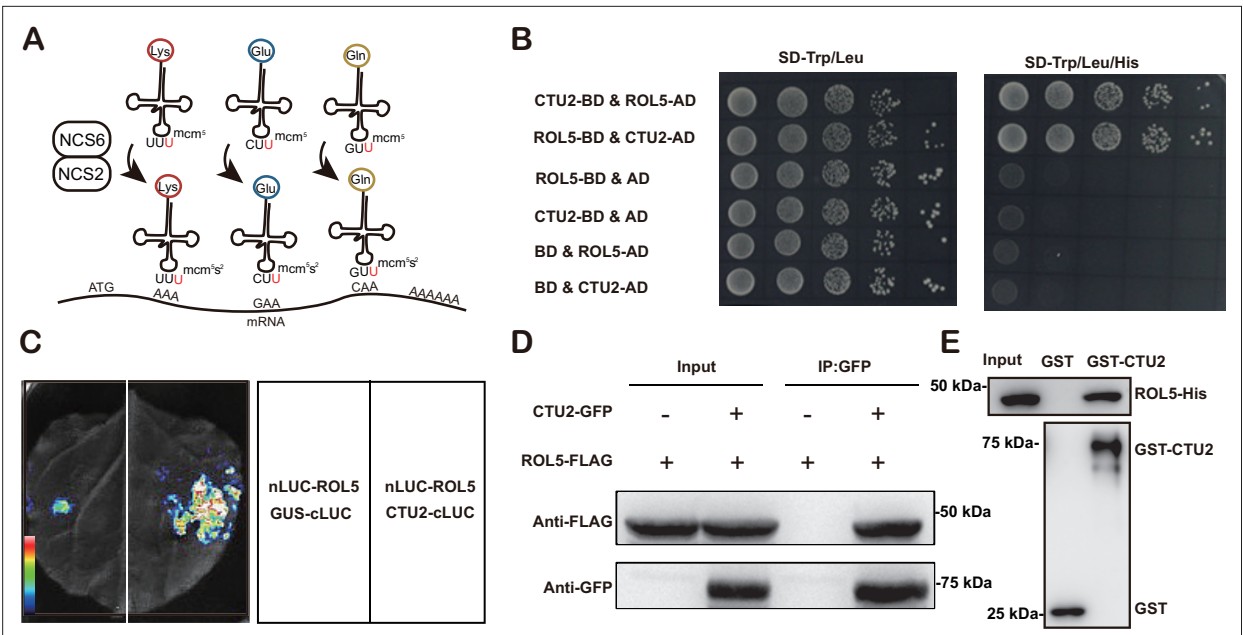

**Figure 2.** ROL5 interacts with CTU2. (**A**) A schematic diagram showing the function of ROL5 and CTU2. The ROL5 homolog NCS6 and the CTU2 homolog NCS2 form a complex to catalyze the mcm⁵s²U modification at wobble nucleotide of tRNA-Lys (UUU), tRNA-Gln (UUC), and tRNA-Glu (UUG), which pair with the AAA, GAA, and CAA codons in mRNA, respectively. (**B**) Yeast two-hybrid assays. The growth of yeast cells on the SD-Trp/Leu/His medium indicates interaction. BD, binding domain. AD, activation domain. (**C**) Split luciferase assays. The indicated proteins were fused to either the C- or N-terminal half of luciferase (cLUC or nLUC) and were transiently expressed in *N. benthamiana*. The luminesce detected by a CCD camera reports interaction. (**D**) Co-immunoprecipitation (CoIP) assays. CTU2-GFP and/or ROL5-FLAG fusion proteins were expressed in *N. benthamiana*. The protein samples were precipitated by GFP-Trap, followed by western blotting using anti-GFP or anti-FLAG antibodies. (**E**) GST pull-down assays. The recombinant GST or GST-CTU2 proteins coupled with glutathione beads were used to pull down His-ROL5, followed by western blotting using anti-His or anti-GST antibodies.

The online version of this article includes the following source data for figure 2:

**Source data 1.** Source data related to *Figure 2D*.

**Source data 2.** Source data related to *Figure 2E*.

## ROL5 interacts with CTU2 in *Arabidopsis*

ROL5 is a homolog of yeast NCS6 (*Leiber et al., 2010*), which forms a protein complex with NCS2 to catalyze mcm⁵s²U34 (*Figure 2A*). The NCS2 homolog in *Arabidopsis* is CTU2 (*Philipp et al., 2014*). To test whether ROL5 interacts with CTU2, we first performed yeast two-hybrid assays. Consistent with the previous finding (*Philipp et al., 2014*), only when ROL5 and CTU2 were co-expressed, the yeasts could grow on the selective medium (*Figure 2B*), indicating that ROL5 interacts with CTU2 in yeast. To test whether they can interact in vivo, we carried out split luciferase assays in *Nicotiana benthamiana*. ROL5 was fused with the N-terminal half of luciferase (nLUC) and CTU2 was fused with the C-terminal half of luciferase (cLUC). An interaction between two proteins brings the two halves of luciferase in close proximity, leading to enzymatic activity and production of luminescence that is detectable using a hypersensitive CCD camera. As shown in *Figure 2C*, the luminescence signal could be detected only when ROL5-nLUC and cLUC-CTU2 were co-expressed. We also performed co-immunoprecipitation (CoIP) assays in *N. benthamiana*. When ROL5-FLAG was co-expressed with CTU2-GFP, ROL5-FLAG could be immunoprecipitated by the GFP-Trap beads (*Figure 2D*). To test whether the interaction is direct, we conducted GST pull-down assays. GST-CTU2 and ROL5-His proteins were expressed in *Escherichia coli* and were purified using affinity resins. As shown in *Figure 2E*, ROL5-His could be specifically pulled down by GST-CTU2, but not by GST alone, suggesting that ROL5 directly interacts with CTU2.

## The tRNA thiolation is required for plant immunity

Given that CTU2 interacts with ROL5, we reasoned that the *ctu2* mutant should have similar phenotypes to *rol5* in response to pathogens. To test this, we infected the T-DNA insertion mutant *ctu2-1*

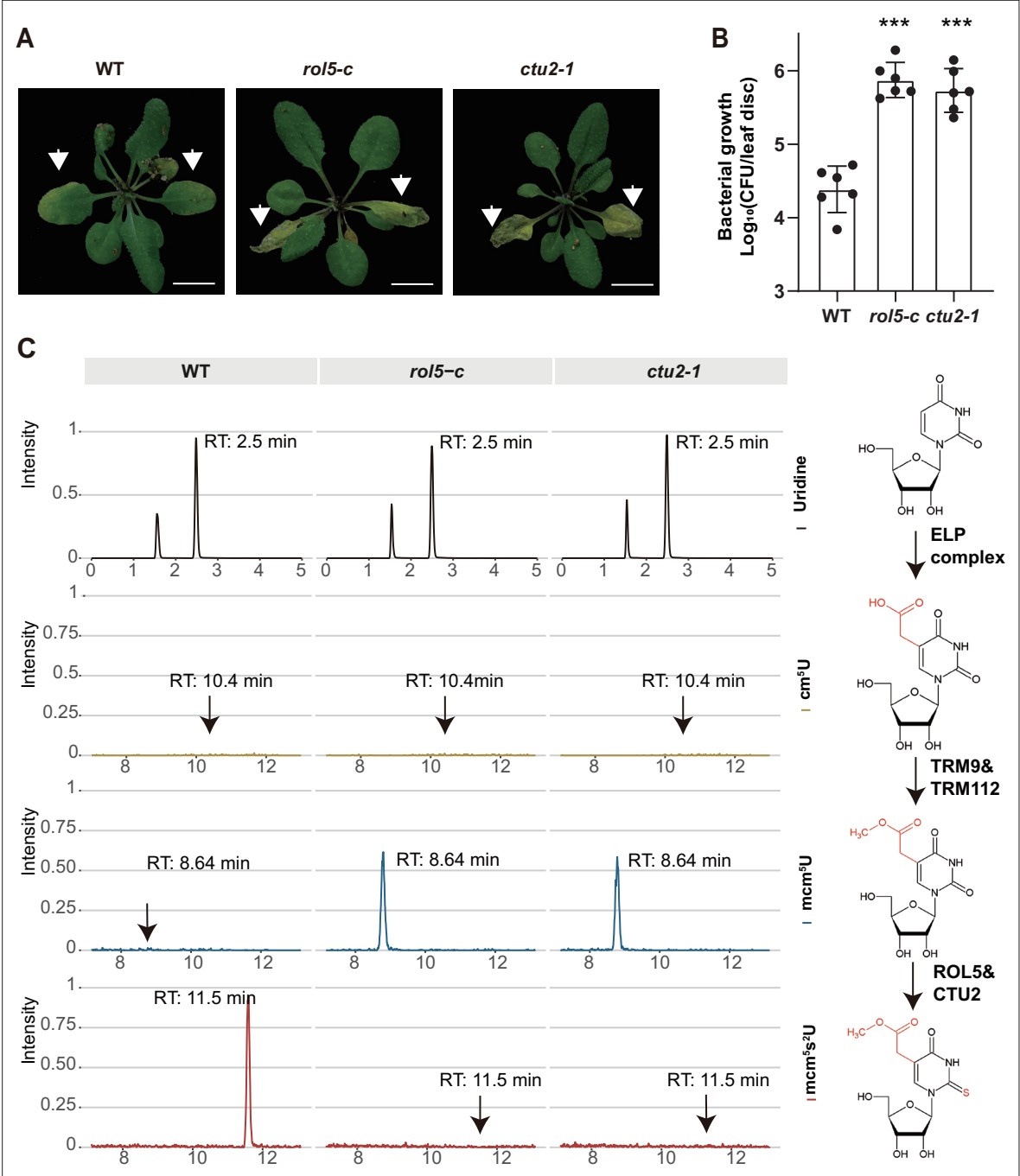

**Figure 3.** ROL5 and CTU2 are required for mcm5s2U modification and plant immunity. (**A and B**) The *rol5-c* and *ctu2-1* mutants are more susceptible to the bacterial pathogen *Psm* ES4326 than wild-type (WT). (**A**) Pictures of *Arabidopsis* plants 3 days after infection. Arrows indicate the leaves inoculated with *Psm* ES4326. Bar = 1 cm. (**B**) The growth of *Psm* ES4326. CFU, colony-forming unit. Error bars represent 95% confidence intervals (n=6). Statistical significance was determined by two-tailed Student's t-test. ***, p<0.001. (**C**) The *rol5-c* and *ctu2-1* mutants lack the mcm5s2U modification. The levels of U, cm5U, mcm5U, and mcm5s2U were quantified through high-performance liquid chromatography coupled with mass spectrometry (HPLC-MS) analyses. The intensity and the retention time of each nucleotide are shown. The structure of each nucleotide and the catalyzing enzymes are shown on the right.

The online version of this article includes the following source data for figure 3:

**Source data 1.** Source data related to *Figure 3A*.

**Source data 2.** Source data related to *Figure 3C*.

with *Psm* ES4326. As expected, the *ctu2-1* mutant is hyper-susceptible to the pathogen (*Figure 3A and B*).

By using N-acryloylamino phenyl mercuric chloride, which binds thiolated tRNAs, previous studies revealed that tRNA thiolation was defective in the *rol5* and *ctu2* mutant (*Leiber et al., 2010*; *Philipp et al., 2014*). To confirm this result, we measured the mcm$^5$U and mcm$^5$s$^2$U levels in WT, *rol5-c*, and *ctu2-1* using high-performance liquid chromatography coupled with mass spectrometry (HPLC-MS). In WT, mcm$^5$U was almost undetectable (*Figure 3C*), indicating that it is efficiently transformed into mcm$^5$s$^2$U in *Arabidopsis*. However, in the *rol5-c* and *ctu2-1* mutants, the mcm$^5$s$^2$U level was undetectable while the mcm$^5$U level was very high, suggesting that both ROL5 and CTU2 are required for mcm$^5$s$^2$U. These data revealed that ROL5 and CTU2 form a complex to catalyze the mcm$^5$s$^2$U modification, which is essential for plant immunity.

## Transcriptome and proteome reprogramming are compromised in *cgb*

To understand why the *cgb* mutant was hyper-susceptible to pathogens, we performed transcriptome and proteome analyses of the *cgb* mutant and the *COM* line. Each sample was divided into two parts, one for transcriptome analysis using RNA sequencing (RNA-seq) approach, and the other for proteome analysis using a tandem mass tag (TMT)-based approach. Principal component analysis showed that the reproducibility between biological replicates was good (*Figure 4—figure supplement 1*). The differentially expressed genes (DEGs) and the differentially expressed proteins (DEPs) between different samples were identified and quantified through data analysis. Regarding the transcriptome, in *COM*, 22% (4819) and 27% (5767) of genes were respectively up-regulated or down-regulated after *Psm* infection (*Figure 4A*). However, only 18% (3986) and 23% (4913) of genes were respectively up-regulated or down-regulated in *cgb*. Regarding the proteome, in *COM*, 16% (1193) and 13% (1021) of proteins were respectively up-regulated or down-regulated after *Psm* infection (*Figure 4B*). In contrast, only 12% (909) and 10% (787) of proteins were respectively up-regulated or down-regulated in *cgb*. Therefore, the numbers of both DEGs and DEPs were reduced in *cgb* compared to those in *COM*.

To further examine the gene expression defects in *cgb*, we compared the expression changes after *Psm* infection between *cgb* and *COM*. Among 4819 up-regulated DEGs in *COM*, the expression changes of 1113 genes were less prominent in *cgb* than in *COM* (*Figure 4C*). These genes were referred to as attenuated genes. Among 1193 up-regulated DEPs in *COM*, the expression changes of 366 proteins were less prominent in *cgb* than in *COM* (*Figure 4D*). These proteins were named attenuated proteins. Gene Ontology (GO) analysis of the attenuated genes and attenuated proteins revealed that many important biological processes were significantly enriched (*Figure 4E and F*). These data suggested that both transcriptome and proteome reprogramming were compromised in *cgb*.

## The translation efficiency of immune-related proteins is compromised in *cgb*

Since the mcm$^5$s$^2$U modification directly regulates translation process (*Nedialkova and Leidel, 2015*; *Schaffrath and Leidel, 2017*), we sought to identify the proteins with compromised translation efficiency. The 366 attenuated proteins in *cgb* may be due to reduced transcription or reduced translation. To distinguish between these two possibilities, we performed Venn diagram analysis between attenuated genes and attenuated proteins, revealing that 261 attenuated proteins were not attenuated at the transcript level, suggesting that the attenuated expression of these proteins is due to reduced translation (*Figure 5A*). GO analysis of these 261 proteins revealed that some immune-related processes (i.e. response to salicylic acid, defense response to bacterium, and immune system process) were significantly enriched (*Figure 5B*). Notably, NPR1 is one of these proteins.

To verify the expression of NPR1, we performed RT-qPCR and western blot analysis. Consistent with transcriptome and proteome data, the transcription levels of NPR1 were similar between *COM* and *cgb* both before and after *Psm* ES4326 infection (*Figure 5—figure supplement 1*), whereas the NPR1 protein level was much higher in *COM* than that in *cgb* after *Psm* ES4326 infection (*Figure 5C*). To further confirm that the translation of NPR1 was reduced in *cgb*, we carried out ribosome profiling experiment. Compared with *COM,* the polysome fractions in *cgb* were reduced (*Figure 5D*),

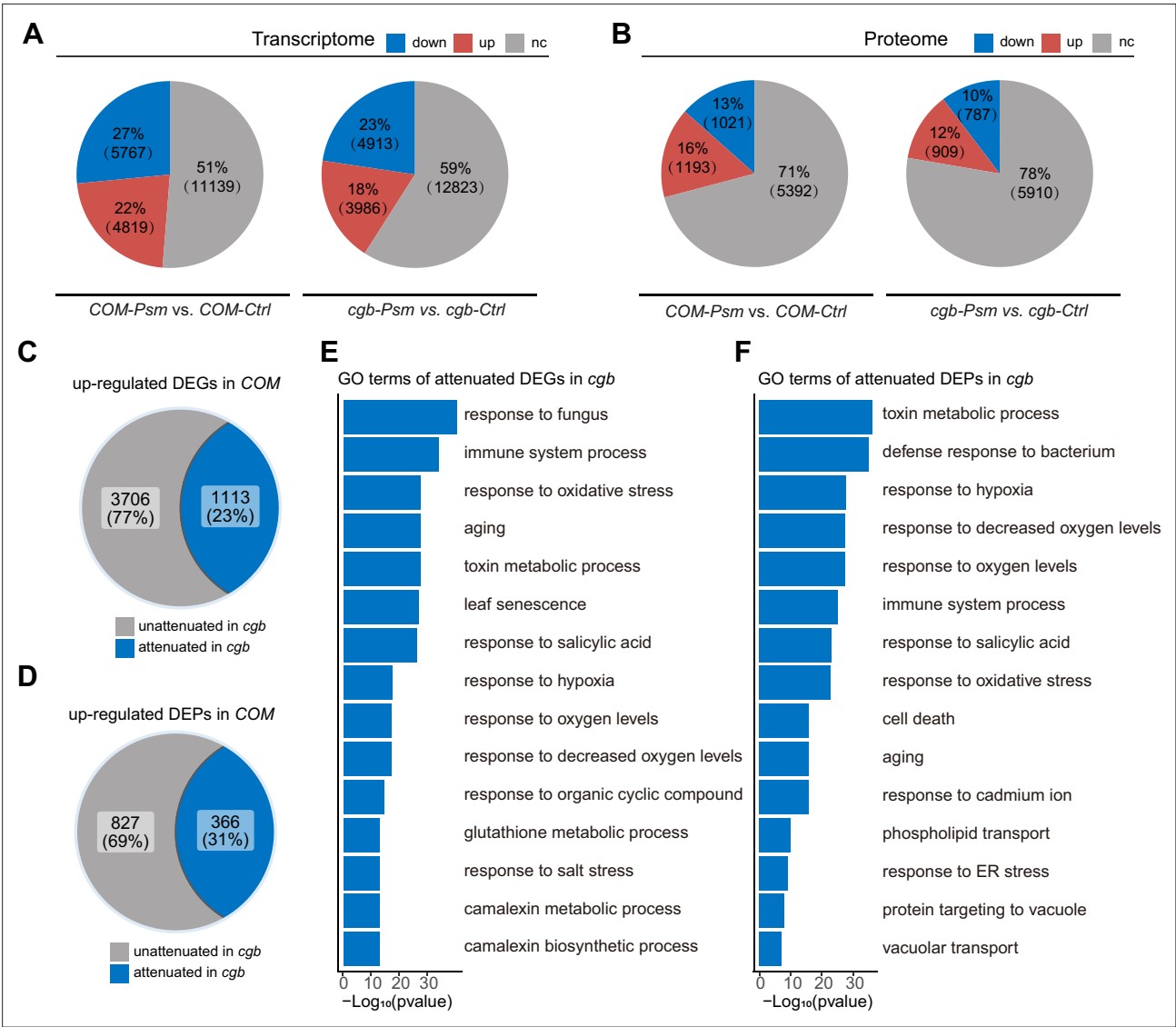

**Figure 4.** The transcriptome and proteome reprogramming are compromised in *cgb*. (**A and B**) The percentage and the number of the differentially expressed genes (DEGs, p-value <0.05, |Log$_2$Foldchange|>Log$_2$1.5, (**A**)) and the differentially expressed proteins (DEPs, p-value <0.05, |Log$_2$Foldchange|>Log$_2$1.2, (**B**)) after *Psm* infection in the *cgb* mutant and the complementation line (*COM*). Down, down-regulated. Up, up-regulated. Nc, no change. (**C and D**) The percentage and the number of the attenuated genes (**C**) and proteins (**D**) in *cgb* among the up-regulated DEGs and DEPs in *COM*. (**E and F**) Gene Ontology (GO) analysis of the attenuated genes (**E**) or proteins (**E**) in *cgb*. The top 15 significantly enriched GO terms are shown.

The online version of this article includes the following source data and figure supplement(s) for figure 4:

**Source data 1.** Source data related to *Figure 4A*.

**Source data 2.** Source data related to *Figure 4B*.

**Source data 3.** Source data related to *Figure 4C*.

**Source data 4.** Source data related to *Figure 4D*.

**Source data 5.** Source data related to *Figure 4E*.

**Source data 6.** Source data related to *Figure 4F*.

**Figure supplement 1.** Principal component analysis (PCA) of the transcriptome (**A**) and proteome samples (**B**).

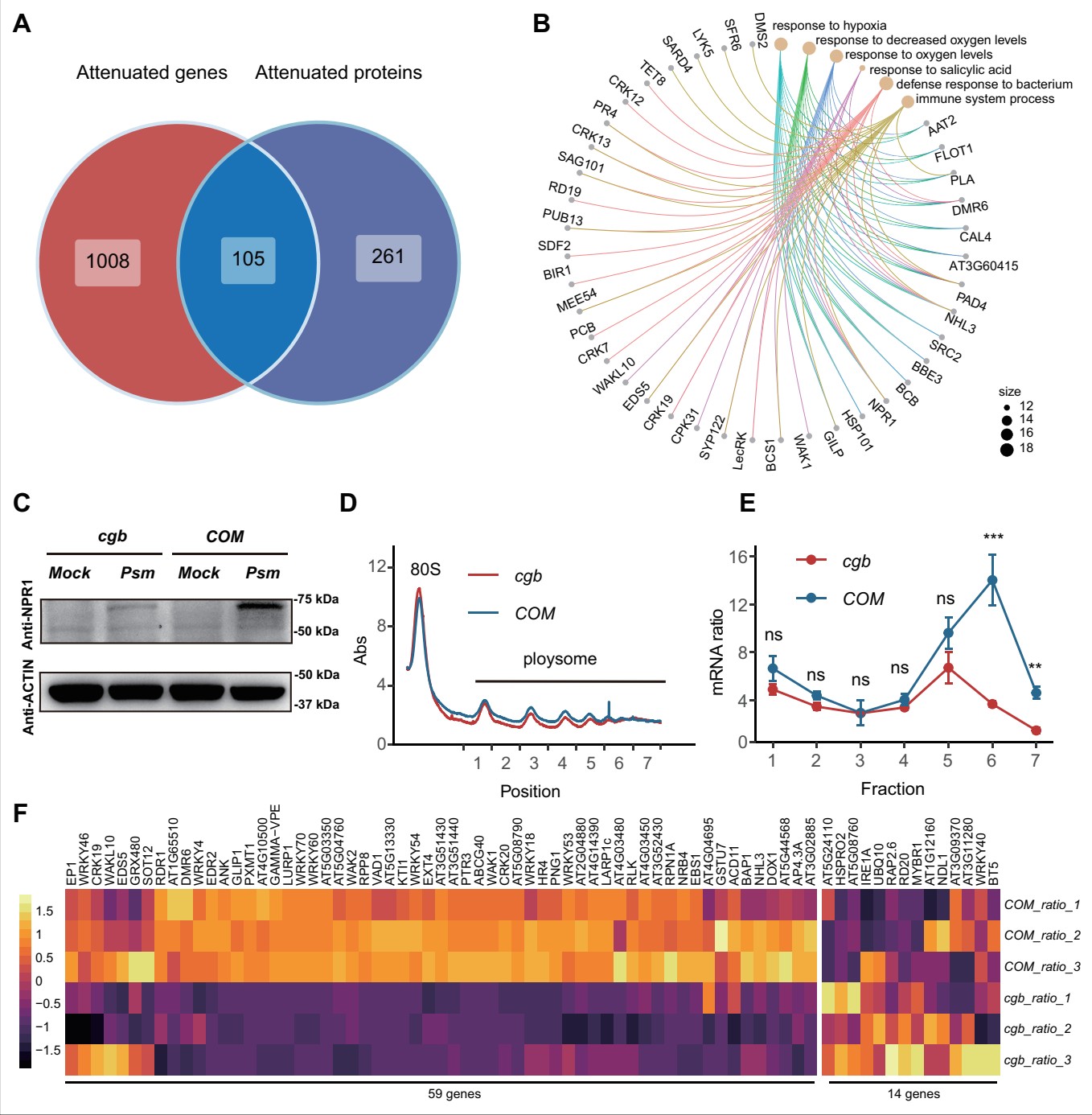

**Figure 5.** The translation of immune-related proteins is compromised in *cgb*. (**A**) Venn diagram analysis of attenuated genes and proteins. (**B**) Gene Ontology (GO) analysis of the 261 attenuated proteins. The top 6 significantly enriched GO terms are shown. (**C**) Western blot analysis of NPR1 protein levels. The 7-day-old seedlings grown on 1/2 MS medium were treated with buffer (10 mM MgCl$_2$, pH 7.5, Mock) or *Psm* ES4326 (OD$_{600}$=0.2) for 48 hr. (**D**) Polysome profiling results. Abs, the absorbance of sucrose gradient at 254 nm. The numbers on the X-axis indicate the polysomal fractions subjected to qPCR analyses. (**E**) The qPCR analyses. The relative mRNA level of *NPR1* in different fractions or in total mRNA was normalized against *UBQ5*. The ratio between the relative mRNA levels in each fraction and in total mRNA was shown (n=3). Statistical significance was determined by two-tailed Student's t-test. **, p<0.01; ***, p<0.001; ns, not significant. (**F**) The heatmap showing the expression changes of salicylic acid (SA)-responsive genes after pathogen infection.

The online version of this article includes the following source data and figure supplement(s) for figure 5:

**Source data 1.** Source data related to *Figure 5A*.

*Figure 5 continued on next page*

*Figure 5 continued*

**Source data 2.** Source data related to *Figure 5B*.

**Source data 3.** Source data related to *Figure 5C*.

**Source data 4.** Source data related to *Figure 5D*.

**Source data 5.** Source data related to *Figure 5E*.

**Source data 6.** Source data related to *Figure 5F*.

**Figure supplement 1.** Analyses of *NPR1* transcript levels in *cgb* and *COM*.

**Figure supplement 2.** The salicylic acid (SA)-mediated protection assay.

**Figure supplement 3.** The genetic relationship between NPR1 and CGB.

suggesting that the overall translation efficiency is lower in *cgb*. As expected, the relative mRNA levels of NPR1 in multiple polysome fractions were significantly lower in *cgb* than in *COM* (*Figure 5E*).

The reduced NPR1 protein level in *cgb* suggested that SA signaling is compromised. To test this possibility, we examined the expression of all the genes (118) belonging to the GO term 'response to salicylic acid'. In our transcriptome data, we could detect the expression of 73 genes, among which 59 genes (80.8%) were reduced in *cgb* compared with *COM* (*Figure 5F*). To further examine the defects of SA signaling in *cgb*, we performed SA-mediated protection assay. The *Arabidopsis* plants were treated with benzothiadiazole (BTH), a functional analog of SA, for 24 hr before infection. As expected, the growth of *Psm* ES4326 was reduced in BTH-treated *COM,* but not *cgb* and *npr1* (*Figure 5—figure supplement 2*). These results suggested that SA signaling is indeed compromised in the *cgb* mutant.

To investigate the genetic relationship between CGB and NPR1, we generated the *cgb npr1* double mutant and examined its disease phenotypes. We found that *cgb npr1* was significantly more susceptible than either *npr1* or *cgb* (*Figure 5—figure supplement 3*). There are two possible reasons for the observed additive effects of *cgb* and *npr1*. First, the translation of *NPR1* was reduced rather than completely blocked in *cgb* (*Figure 5C*). In other words, NPR1 still has some function in *cgb*. But in the *cgb npr1* double mutant, the function of NPR1 is completely abolished, which explains why *cgb npr1* was more susceptible than *cgb*. Second, in addition to NPR1, some other immune regulators (such as PAD4, EDS5, and SAG101) were also compromised in *cgb* (*Figure 5B*), which explains why *cgb npr1* was more susceptible than *npr1*.

## Discussion

Upon pathogen infections, plants need to efficiently reprogram their gene expression, allowing the transition from growth to defense. However, how translation contributes to the immune response is not well studied. It is known that tRNA thiolation is required for efficient protein expression (*Nedialkova and Leidel, 2015*; *Schaffrath and Leidel, 2017*). Here, we show that tRNA thiolation is abolished in the *cgb* mutant (*Figure 3*), leading to disease hyper-susceptibility (*Figure 1*). We found that the translation of many immune-related proteins was reduced in *cgb* (*Figure 5*). Therefore, our study strongly suggested that tRNA thiolation is required for plant immunity, revealing an additional mechanism underlying plant immune responses. It is possible that tRNA thiolation is a regulatory step during immune responses. However, since many defense-related proteins are up-regulated after pathogen infection (*Figure 4B*), we cannot rule out the possibility that tRNA thiolation just becomes a limiting factor due to the high demand of translation resource during immune responses. More studies are required to distinguish these two possibilities.

The SA receptor NPR1 is the master regulator of SA signaling. NPR1 can function as a transcription coactivator to regulate gene expression and an adaptor of ubiquitin E3 ligase to mediate protein degradation (*Yu et al., 2022*; *Yu et al., 2021*; *Zavaliev et al., 2020*). It has been shown that the activity of NPR1 is regulated at multiple levels including post-translational modifications such as phosphorylation, ubiquitination, S-nitrosylation, and sumoylation (*Saleh et al., 2015*; *Spoel et al., 2009*; *Tada et al., 2008*). However, how NPR1 is regulated at the translational level is unknown. Here, we show that the tRNA thiolation-mediated translation control is required for the optimal expression of NPR1 (*Figure 5B and D*), revealing an additional layer of regulation for NPR1.

The tRNA thiolation modification is highly conserved in eukaryotes. However, its biological functions in plants are less well understood. Previously, it was reported that tRNA thiolation regulates the development of root hairs, chloroplasts, and leaf cells (*Leiber et al., 2010*; *Philipp et al., 2014*). Recently, it was found that tRNA thiolation is required for heat stress tolerance (*Xu et al., 2020*). Our study revealed an additional biological function of tRNA thiolation in plant immunity. It will also be interesting to test whether tRNA thiolation is required for responses to other stresses such as drought, salinity, and cold.

The ELP complex is composed of six proteins, with ELP1, ELP2, and ELP3 forming a core subcomplex, and ELP4, ELP5, and ELP6 forming an accessory sub-complex. The ELP complex catalyzes the $cm^5U$ modification, which is the precursor of $mcm^5s^2U$ catalyzed by ROL5 and CTU2. As expected, the $mcm^5s^2U$ modification was undetectable in the *elp* mutants such as *elp3* and *elp6* mutants (*Leitner et al., 2015*; *Mehlgarten et al., 2010*). Interestingly, similar to the *rol5* and *ctu2* mutants, the *elp2* and *elp3* mutants were hyper-susceptible to pathogens (*DeFraia et al., 2010*; *Defraia et al., 2013*; *Wang et al., 2013*). In addition to tRNA modification, the ELP complex has several other distinct activities including histone acetylation, α-tubulin acetylation, and DNA demethylation (*Wang et al., 2013*). Therefore, it is difficult to dissect which activity of the ELP complex contributes to plant immunity. However, the only known activity of ROL5 and CTU2 is to catalyze tRNA thiolation. Considering that the *elp*, *rol5*, and *ctu2* mutants are all defective in tRNA thiolation, it is likely the tRNA modification activity of the ELP complex underlies its function in plant immunity.

Previous studies have identified numerous pathogen-responsive genes through transcriptome analysis (*Zhang et al., 2020*). However, the correlation between mRNAs and proteins is not always that strong (*Lahtvee et al., 2017*; *Schwanhäusser et al., 2011*). Given that proteins are major players in cellular functions, it is necessary to study immune responses at the protein level. Through high-throughput proteome analysis, we found 2215 proteins differentially accumulated after *Psm* infection in *Arabidopsis* (*Figure 4*). To our knowledge, this is the largest dataset of pathogen-responsive proteins in *Arabidopsis*. We believe that this dataset will provide a good research resource for future studies on plant immunity.

# Materials and methods

**Key resources table**

| Reagent type (species) or resource | Designation | Source or reference | Identifiers | Additional information |
|---|---|---|---|---|
| Gene (*Arabidopsis thaliana*) | *ROL5* | TAIR | AT2G44270 | |
| Gene (*Arabidopsis thaliana*) | *CTU2* | TAIR | AT4G35910 | |
| Genetic reagent (*Arabidopsis thaliana*) | *cgb* | This paper | | It contains a T-DNA insertion in the fourth exon of ROL5 and is hypersusceptible to pathogen. |
| Genetic reagent (*Arabidopsis thaliana*) | *COM* | This paper | | It contains the coding sequence of ROL5 driven by 35S promoter in cgb. |
| Genetic reagent (*Arabidopsis thaliana*) | *rol5-c* | This paper | | The mutant was generated using CRISPR-Cas9 system. It contains a 2-bp deletion in the first exon of ROL5. |
| Genetic reagent (*Arabidopsis thaliana*) | *ctu2-1* | ABRC | SALK_032692 | |
| Genetic reagent (*Arabidopsis thaliana*) | *npr1-1* | *Cao et al., 1997* | | |
| Strain, strain Background (*Escherichia coli*) | *BL21* | TransGen | Cat # CD901-02 | Electrocompetent cells |
| Strain, strain background (*Escherichia coli*) | *DH5α* | TransGen | Cat # CD201-01 | Electrocompetent cells |
| Strain, strain background (*Agrobacterium tumefaciens*) | *GV3101* | Sangon | Cat # B528430 | Electrocompetent cells |
| Strain, strain background (*Saccharomyces cerevisiae*) | *AH109* | Clontech | Cat # 630489 | Electrocompetent cells |

*Continued on next page*

*Continued*

| Reagent type (species) or resource | Designation | Source or reference | Identifiers | Additional information |
|---|---|---|---|---|
| Strain, strain background (*Pseudomonas syringae* pv. *Maculicola*) *Psm 4326* | | *Durrant et al., 2007* | ES4326 | |
| Antibody | Anti-NPR1 (Rabbit polyclonal) | From Dr. Li Yang | | WB(1:3000) |
| Antibody | Anti-His (Mouse monoclonal) | Abclonal | Cat # AE003 | WB(1:5000) |
| Antibody | Anti-GST (Mouse monoclonal) | Abclonal | Cat # AE001 | WB(1:5000) |
| Antibody | Anti-FLAG (Mouse monoclonal) | Promoter | | WB(1:5000) |
| Antibody | Anti-GFP (Mouse monoclonal) | Promoter | | WB(1:5000) |
| Other | GFP-Trap | chromotek | Cat # gtma | |
| Other | Hypersil GOLD | Thermo Fisher | Cat # 25005-254630 | |

## Plant material and growth conditions

All *Arabidopsis* seeds used in this study are in *Columbia-0* background. The *npr1-1* mutant was described previously (*Cao et al., 1997*). The *cgb* mutant and the complementation line were generated in this study. The mutant of *ctu2-1* (SALK_032692) was purchased from ABRC. The *rol5-c* mutant was generated using EC1-based CRISPR-Cas9 system (*Wang et al., 2015*). All seeds were sterilized with 2% Plant Preservative Mixture-100 (Plant Cell Technology) at 4°C in the dark for 2 days and then were plated on Murashige and Skoog (MS) medium with 1% sucrose and 0.3% phytagel. The plants were grown under long-day conditions at 22°C (16 hr of light/8 hr of dark; supplied by white-light tubes).

## Strains and growth conditions

*E. coli* strain *DH5α* for molecular cloning was cultured in LB medium at 37°C. *E. coli* strain *BL21* (DE3) for recombinant protein expression was cultured in LB medium at 16°C. *Agrobacterium tumefaciens* strain GV3101 for transformation was cultured in Yeast Extract Beef (YEB) medium at 28°C. *Psm* ES4326 for infection assay was cultured in King's B (KB) medium at 28°C. Yeast strain AH109 for yeast two-hybrid assay was cultured in Yeast Peptone Dextrose (YPD) medium or SD medium at 28°C.

## Vector constructions

The vectors were constructed using the digestion-ligation method or a lighting cloning system (Biodragon Immunotechnology). For complementation experiment, *ROL5* was inserted into *Nco* I/*Xba* I-digested *pFGC5941*. For pull-down assays, *CTU2* was inserted into *Bam*H I/*Xho* I-digested *pGEX-6P-1*; *ROL5* was inserted into *Nco* I/*Hin*d III-digested pET28a. For split luciferase assays, *ROL5* and *CTU2* were cloned into the *Kpn* I/*Sal* I-digested *pJW771* and *pJW772*, respectively. For yeast two-hybrid assays, *ROL5* and *CTU2* were cloned into *EcoR* I/*BamH* I-digested *pGBKT7* and *pGADT7*. For CoIP assays, *ROL5-FLAG* and *CTU2-GFP* were cloned into *Nco* I/*Xba* I-digested *pFGC5941*. To generate *rol5-c*, the target sequence was designed and cloned into *pHEE401* as described previously (*Wang et al., 2015*). The primer sequences used for cloning are listed in *Appendix 1—table 1*.

## Reverse transcription and qPCR

The total RNA or the RNA in ribosome fractions was extracted using TRIzol Reagent (Invitrogen). The cDNA was synthesized using HiScript II Q RT SuperMix (Vazyme). The qPCR analyses were performed using the AceQ qPCR SYBR Green Master Mix (Vazyme). *UBQ5* was used as the internal reference gene. Primers used for qPCR are listed in *Appendix 1—table 1*.

## Pathogen infection

The third and fourth leaves of 3-week-old *Arabidopsis* plants were infiltrated with *Psm* ES4326 ($OD_{600}$=0.0002) using a needleless syringe. Three days after infection, the leaves were sampled to

measure the growth of *Psm* ES4326 as described previously (*Durrant et al., 2007*). For SA-mediated protection assay, the 3-week-old *Arabidopsis* plants were treated with 600 μM BTH (Syngenta) for 24 hr before infection.

## Yeast two-hybrid assays

Matchmaker GAL4 Two-Hybrid System (Clontech) was used and the assays were performed according to the user manual. Briefly, the bait (in *pGBKT7*) and prey (in *pGADT7*) vectors were co-transformed into the yeast strain AH109. The protein interactions were determined by yeast growth on SD/-Leu/-Trp/-His/ medium. The empty vectors were used as negative controls.

## In vitro pull-down assays

The GST pull-down assays were performed as previously described (*Chen et al., 2023a* , *Chen et al., 2023b*). Briefly, ROL5-His, GST, and GST-CTU2 proteins were expressed in *E. coli* BL21 (DE3). GST (5 μg) and GST-CTU2 (5 μg) were coupled to glutathione beads (GE Healthcare Life Sciences) and then were incubated with ROL5-His (10 μg) in 0.5 mL binding buffer (50 mM Tris-HCl pH 7.5, 150 mM NaCl, 1 mM EDTA, and 2 mM DTT) at 4°C for 2 hr. The beads were washed three times with washing buffer (binding buffer plus 2% NP-40), boiled in 1× SDS loading buffer, and analyzed by western blot using anti-GST (Abclonal) or anti-His (Abclonal) antibodies.

## CoIP assays

The CoIP assays were performed as previously described (*Chen et al., 2021*). 35S:*ROL5-FLAG* and 35S:*CTU2-GFP* were transformed into *A. tumefaciens* GV3101. 35S:*ROL5-FLAG* strain ($OD_{600}$=1) was mixed with the same volume of buffer or 35S:*CTU2-GFP* strain ($OD_{600}$=1) and was infiltrated into *N. benthamiana* leaves. After 48 hr, the infiltrated leaves were ground in liquid nitrogen and were resuspended in IP buffer (20 mM Tris-HCl pH 7.5, 50 mM NaCl, 1 mM EDTA, 0.1% SDS, 1% Triton X-100, 1 mM PMSF, 100 μM MG132, 1× protease inhibitor cocktail) for total protein extraction. The lysates were incubated with GFP-Trap magnetic beads (Chromotek) at 4°C for 2 hr. The beads were washed using washing buffer (20 mM Tris-HCl pH 7.5, 150–500 mM NaCl, 1 mM EDTA, 1 mM PMSF, 1× Protease Inhibitor Cocktail) and then boiled in 1× SDS loading buffer. The western blotting was performed using anti-FLAG (Promoter) and anti-GFP (Promoter) antibodies.

## Split luciferase assays

Split luciferase assay was performed as described previously (*Chen et al., 2008*). The constructs were transformed into *A. tumefaciens* strain GV3101 ($OD_{600}$=1). The resultant strains were then infiltrated into leaves of *N. benthamiana*. After 48 hr, 1 mM luciferin (GOLDBIO) was applied onto leaves and the images were captured using Lumazone imaging system equipped with 2048B CCD camera (Roper).

## Quantification of tRNA modifications

Quantification of tRNA modifications was performed using liquid chromatography coupled with mass spectrometry according to a previous study (*Su et al., 2014*). Total tRNA was extracted using a microRNA kit (Omega Bio-Tek). Five micrograms of tRNA were hydrolyzed in 10 μL enzymic buffer (1 U benzonase, 0.02 U phosphodiesterase I, and 0.02 U alkaline phosphatase) at 37°C for 3 hr. The UHPLC system (Thermo Fisher Scientific) coupled with TSQ Altis Triple Quadrupole Mass Spectrometer (Thermo Fisher Scientific) was used for quantification of tRNA modification. For the liquid chromatography, the Hypersil GOLD HPLC column (3 μm, 150×2.1 mm²; Thermo Fisher Scientific) was used. The solvent gradient was set as the protocol (*Su et al., 2014*). The Tracefinder software (Thermo Fisher Scientific) was further used for peak assignment, area calculation, and normalization. Corresponding structures and molecular masses were obtained from the Modomics database (https://iimcb.genesilico.pl/modomics/modifications).

## RNA and protein extraction for transcriptome and proteome analysis

The samples were ground in liquid nitrogen and divided into two parts, one for transcriptome analysis and the other for proteome analysis. Total RNA was extracted using TRIzol Reagent (Invitrogen). Library preparation and RNA-sequencing were performed by Novogene Cooperation. Total proteins

were extracted using phenol-methanol method (*Deng et al., 2007*). The protein concentration was determined with 2D Quant Kit (GE Healthcare Life Sciences) using bovine serum albumin as a standard.

## Proteome analysis

For trypsin digestion, 60 µg proteins of each sample were reduced with 20 mM Tris-phosphine for 60 min at 30°C. Cysteines were alkylated with 30 mM iodoacetamide for 30 min at room temperature in the dark. Proteins were precipitated with 6 volumes of cold acetone overnight and then dissolved in 50 mM triethylammonium bicarbonate (TEAB). Proteins were digested with trypsin (protease/protein = 1/25, wt/wt) overnight at 37°C.

For TMT labeling, each sample containing 25 µg of peptide in 50 mM TEAB buffer was combined with its respective 10-plex TMT reagent (Thermo Fisher Scientific) and incubated for 1 hr at room temperature. Three biological replicates were labeled respectively for each sample, in which *COM* was labeled with 126-, 127N-, and 128C- of the 10-plex TMT reagent, while *cgb* was labeled with 129N-, 130C-, and 131- of the 10-plex TMT reagents. The labeling reactions were stopped by the addition of 2 µL of 5% hydroxylamine.

For LC-MS/MS analysis, multiplexed TMT-labeled samples were combined, vacuum dried, reconstituted in 2% acetonitrile and 5 mM ammonium hydroxide (pH 9.5), and separated with the Waters Acquity BEH column (C18, 1.7 µm, 100 mm, Waters) using UPLC system (Waters) at a flow rate of 300 µL/min. Total of 24 fractions were collected, combined into 12 fractions, and vacuum dried for LC-MS/MS analysis. The solvent gradient was set as previously described (*Deng et al., 2007*). Samples were then analyzed on an Ultimate 3000 nano UHPLC system (Thermo Fisher Scientific) coupled online to a Q Exactive HF mass spectrometer (Thermo Fisher Scientific). The trapping column (PepMap C18, 100 Å, 100 µm×2 cm, 5 µm) and an analytical column (PepMap C18, 100 Å, 75 µm i.d.×50 cm long, 2 µm) were used for separation of the samples. The solvent gradient and MASS parameters were set as previously described (*Deng et al., 2007*).

## Transcriptome data analysis

Raw reads were processed and aligned to the *Arabidopsis* genome (https://www.arabidopsis.org) using STAR (v.2.6.1a). Genes with over 10 reads were filtered and processed using DESeq2 (v.1.22.2) to identify the DEGs (p-value <0.05, $|Log_2FoldChange|>Log_21.5$) (*Love et al., 2014*).

## Proteome data analysis

Raw data were processed using Proteome Discoverer (v.2.2.0.388) and aligned to *Arabidopsis* genome (https://www.arabidopsis.org) with the SEQUEST HT search engine. Searches were configured with static modifications for the TMT reagents (+229.163 Da). The precursor mass tolerance was set as 10 ppm; the fragment mass tolerance was set as 0.02 Da; the trypsin missed cleavage was set as 2. The reversed sequence decoy strategy was used to control peptide false discovery. The peptides with q scores <0.01 were accepted, and at least one unique peptide was required for matching a protein entry for its identification. PSMs (peptide spectrum matches) results were processed with DESeq2 (v.1.22.2) to identify the DEPs (p-value <0.05, $|Log_2FoldChange|>Log_21.2$).

## GO and heatmap analysis

The DEGs or proteins were analyzed by using Clusterprofile (v.3.18.1) (*Yu et al., 2012*). The heatmap analysis was processed by using pheatmap2 (v.1.0.12).

## Analysis of NPR1 protein level

The seedlings were ground in liquid nitrogen and were resuspended in lysis buffer (50 mM Tris-HCl, pH 7.4, 150 mM NaCl, 1% Triton X-100, 1% sodium deoxycholate, 0.1% SDS, 200 mM DTT, 1 mM PSMF, 50 µM MG132, 1× protease inhibitor cocktail). After centrifuging, the supernatants were mixed with the same volume of 2× SDS loading buffer and were incubated at 75°C for 15 min. The western blotting was performed using an anti-NPR1 antibody (provided by Li Yang from China Agricultural University).

## Ribosome profiling

The ribosome profiling was performed as previously described with some modifications (*Hsu et al., 2016*; *Xu et al., 2017*). The plant sample (0.05–0.1 g) was ground in liquid nitrogen and extracted

with 1 mL ribosome lysis buffer (200 mM Tris-HCl pH 8.0, 200 mM KCl, 35 mM MgCl$_2$, 1% Triton X-100, 100 µM MG132, 1 mM DTT, and 100 µg/mL cycloheximide), followed by ultracentrifugation at 4°C for 2 hr (38,000 rpm, Beckman, SW41 rotor) through a 20–60% sucrose gradient (40 mM Tris-HCl, pH 8.4, 20 mM KCl, 10 mM MgCl$_2$, and 50 µg/mL cycloheximide) prepared by Gradient Master (Biocomp Instruments). The profiling signals were recorded by Piston Gradient Fractionator (Biocomp Instruments).

## Acknowledgements

We are grateful to Dr. Pascal Genschik for critical revision, Dr. Zhipeng Zhou for helpful discussion, Dr. Peng Chen for technical support, and Dr. Li Yang for providing the anti-NPR1 antibody. This work is supported by the National Natural Science Foundation of China (31970311, 32000373, and 32270306), HZAU-AGIS Cooperation Fund (SZYJY2022004), and BaiChuan Program.

## Additional information

### Funding

| Funder | Grant reference number | Author |
|---|---|---|
| National Natural Science Foundation of China | 31970311 | Shunping Yan |
| HZAU-AGIS Cooperation Fund | SZYJY2022004 | Shunping Yan |
| National Natural Science Foundation of China | 32000373 | Xiaodan Yu |
| National Natural Science Foundation of China | 32270306 | Shunping Yan |

The funders had no role in study design, data collection and interpretation, or the decision to submit the work for publication.

### Author contributions

Xueao Zheng, Conceptualization, Investigation, Writing – original draft, Project administration, Writing – review and editing; Hanchen Chen, Investigation, Project administration, Writing – review and editing; Zhiping Deng, Formal analysis, Project administration; Yujing Wu, Linlin Zhong, Chong Wu, Investigation; Xiaodan Yu, Funding acquisition, Writing – review and editing; Qiansi Chen, Writing – review and editing; Shunping Yan, Conceptualization, Supervision, Funding acquisition, Writing – original draft, Writing – review and editing

### Author ORCIDs

Xueao Zheng http://orcid.org/0000-0001-6204-7611
Hanchen Chen http://orcid.org/0009-0004-5323-6989
Zhiping Deng http://orcid.org/0000-0001-9663-3088
Linlin Zhong http://orcid.org/0000-0003-1908-1413
Qiansi Chen http://orcid.org/0000-0003-2800-5703
Shunping Yan http://orcid.org/0000-0002-3665-1310

### Decision letter and Author response

Decision letter https://doi.org/10.7554/eLife.93517.sa1
Author response https://doi.org/10.7554/eLife.93517.sa2

## Additional files

### Supplementary files
• MDAR checklist

## Data availability

RNA sequencing datasets have been deposited to GSE database with an accession number GSE183087. The mass spectrometry proteomics data have been deposited to the ProteomeXchange Consortium via the iProX partner repository with the dataset identifier PXD028189. Data analysis scripts are available on GitHub (copy archived at *Zheng, 2022*). Source data files have been provided for Figures 1B, 1D, 1E, 2D, 2E, 3B, 3C, Figure 4, and Figure 5.

The following datasets were generated:

| Author(s) | Year | Dataset title | Dataset URL | Database and Identifier |
|---|---|---|---|---|
| Zheng X, Wu C, Yan S | 2022 | The thiolation modification of tRNA is essential for plant immunity | https://www.ncbi.nlm.nih.gov/geo/query/acc.cgi?acc=GSE183087 | NCBI Gene Expression Omnibus, GSE183087 |
| Zheng X, Yan S | 2021 | The thiolation modification of tRNA is essential for plant immunity | https://proteomecentral.proteomexchange.org/cgi/GetDataset?ID=PXD028189 | ProteomeXchange, PXD028189 |

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

# Appendix 1

**Appendix 1—table 1.** The primers used in this study.

| Name | Sequence ( 5'–3' ) | Application |
|---|---|---|
| ROL5-F1 | ACATTACAATTACATTTACAATTACATGGAGGCCAAGAACAAGAA | For complementation |
| ROL5-R1 | GGGTCTTAATTAACTCTCTAGATTAGAAATCCAGAGATCCACAT | |
| ROL5-F2 | CGGAATTC ATGGAGGCCAAGAACAAGA | For Y2H |
| ROL5-R2 | CGGGATCC TTAGAAATCCAGAGATCCAC | |
| CTU2-F1 | CGGAATTC ATGGCTTGTAATTCCTCAG | |
| CTU2-R1 | CGGGATCC TTAGACAACCTCTTCATCGT | |
| ROL5-F3 | GGGGTACCATGGAGGCCAAGAACAAGA | For split luc |
| ROL5-R3 | GCGTCGACGAAATCCAGAGATCCAC | |
| CTU2-F2 | GGGGTACCATGGCTTGTAATTCCTCAG | |
| CTU2-R2 | GCGTCGACTTAGACAACCTCTTCATCGT | |
| GUS-F | acgcgtcccggggcggtaccATGGTAGATCTGAGGGTAAA | |
| GUS-R | cgaaagctctgcaggtcgacCTATTGTTTGCCTCCCTGCTG | |
| ROL5-F0 | TGACTGCTCCCTACCTGTCGAGTTTTAGAGCTAGAAATAGC | For CRISPR mutant of ROL5 |
| ROL5-R0 | AACGAGACGTCCCGTCCTCAAACAATCTCTTAGTCGACTCTAC | |
| ROL5-BsF | ATATATGGTCTCGATTGACTGCTCCCTACCTGTCGAGTT | |
| ROL5-BsR | ATTATTGGTCTCGAAACGAGACGTCCCGTCCTCAAACAA | |
| ROL5-F4 | TTGAAAGGTTTACATCTTGGAAT | For sequencing of target sites |
| ROL5-R4 | AAAGGTGATTGCTTAGATTCTGATT | |
| ROL5-F5 | CTCAAAAACCTCATAAAAGCACTCT | |
| ROL5-R5 | AACTGCGTCACTGTCTTTACTCT | |
| ROL5-F6 | TTAAGAAGGAGATATACCATGGGCATGGAGGCCAAGAACAAGA | For protein expression |
| ROL5-R6 | GAGTGCGGCCGCAAGCTTTTAGAAATCCAGAGATCCAC | |
| CTU2-F3 | TTCCAGGGGCCCCTGGGATCCATGGCTTGTAATTCCTCAG | |
| CTU2-R3 | AGTCACGATGCGGCCGCTCGAGTTAGACAACCTCTTCATCGT | |
| ROL5-F7 | CAATTACATTTACAATTACATGGAGGCCAAGAACAAGA | For co-immunoprecipitation |
| ROL5-R7 | GGGTCTTAATTAACTCTCTAGATTTGTCATCATCGTCTTTG | |
| CTU2-F4 | CAATTACATTTACAATTACATGGCTTGTAATTCCTCAGG | |
| CTU2-R4 | GGGTCTTAATTAACTCTCTAGATTACTTGTACAGCTCGTCCA | |
| cgb-LP | GTATGAGAAGTGATTGAGTATGTG | For genotyping |
| cgb-RP | TCGATGTGCACCTACTTAATCTAC | |
| cgb-RB | CTAATGAGTGAGCTAACTCAC | |
| ctu2-LP | TCACATTGCATTGAATCATCC | For genotyping |
| ctu2-RP | TCAAATTTAGCACATGGGACC | |

*Appendix 1—table 1 Continued on next page*

*Appendix 1—table 1 Continued*

| Name | Sequence ( 5'–3' ) | Application |
|---|---|---|
| ROL5-F1 | GGAGCTGCGTTATTGAAAGTAG | |
| ROL5-R1 | CCACGATATGCATTAGGAGAGT | |
| UBQ5-F1 | GAAGATCCAAGACAAGGAAGGA | For qPCR |
| UBQ5-R1 | CTTCTTCCTCTTCTTAGCACCA | |
| NPR1-P1 | ATGATTTCTACAGCGACGCTAA | |
| NPR1-P2 | GACTTCGTAATCCTTGGCAATC | |

