## [Editor Report]

This valuable study provides solid evidence for a role of tRNA thiolation in Arabidopsis immunity through genetic, transcriptomic, and proteomic approaches, specifically that the tRNA mcm5s2U modification affects SA signaling through NPR1 translation.

---

## [Decision Letter]

[Editors' note: this paper was reviewed by Review Commons.]

---

## [Author Response]

Point-by-point description of the revisions

Reviewer #1 (Evidence, reproducibility and clarity (Required)):Review commentThe article titled "The tRNA thiolation-mediated translational control is essential for plant immunity" by Zheng et al. highlights the critical role of tRNA thiolation in Arabidopsis plant immunity through comprehensive analysis, including genetics, transcriptional, translational, and proteomic approaches. Through their investigation, the authors identified a cbp mutant, resulting in the knockout of ROL5, and discovered that ROL5 and CTU2 form a complex responsible for catalyzing the mcm5s2U modification, which plays a pivotal role in immune regulation. The findings from this study unveil a novel regulatory mechanism for plant defense. Undoubtedly, this discovery is innovative and holds significant potential impact. However, before considering publication, it is necessary for the authors to address the various questions raised in the manuscript concerning the experiments and analysis to ensure the reliability of the study's conclusions.

Thank you very much for your support and suggestions!

Here is Comments:Line 64-65:The author mentioned that 'While NPR1 is a positive regulator of SA signaling, NPR3 and NPR4 are negative regulators.' However, several recent discoveries are suggesting that it may not be a definitive fact that NPR3 and NPR4 are negative regulators. Therefore, I recommend the authors to review this section in light of the findings from recent papers and make necessary edits to reflect the most current understanding.

Thank you for your feedback. Since we mainly focused on NPR1 in this study, we removed this sentence to avoid confusion. We provided additional information about NPR1 in the Introduction section to emphasize the importance of NPR1 (Line 64-68).

Line 182- & Figure 4:The author conducted RNA-seq, Ribo-seq, and proteome analysis. Describing the analysis as "transcriptional and translational" using RNA-seq and proteome data seems not entirely accurate. Proteome data compared with RNA-seq not only reflects translational changes but may also encompass post-translational regulations that contribute to the observed differences. To maintain precision, the title of this section should either be modified to "transcriptional and protein analysis" or, alternatively, compare RNA-seq and Ribo-seq data to demonstrate the transcriptional and translational changes more explicitly.

Thank you for your suggestions. We agree with you and thus change the description accordingly throughout the manuscript.

Line 229-235 and Figure 5C:The interpretation of Figure 5C's polysome profiling results is inconclusive. There does not seem to be a noticeable difference in polysomal fractions between the cab mutant and CAM. The observed differences in the overlay of multiple polysome fractions between cab and COM could be primarily influenced by baseline variations rather than a significant decrease in the polynomial fractions in cpg. Therefore, it is necessary to carefully review other relevant papers that discuss polysome fraction data and their analysis. By doing so, the authors can make the appropriate corrections to ensure accurate interpretations.

Thank you for your comments. We agree that the difference between *cgb* and *COM* was not dramatic visually. This is a common feature of polysome profiling assay (e.g. Extended Data Figure 1f in Nature 545: 487–490; Figure 1c in Nature Plants, 9: 289–301). In our case, the difference between polysome fractions was unlikely due to the baseline variation for two reasons. First, baseline variation affects monosome and polysome fractions in the same way. However, our results showed the monosome fraction of *cgb* is higher than that of *COM*, whereas the polysome fraction of *cgb* is lower than that of *COM*. Second, this result was repeatedly detected. For better visualization, we adjusted the scale of Y axis in the revised manuscript (Figure 5D).

Line 482 Ion Leakage assay:I could not find the ion leakage assay in this manuscript, so I wonder why it is mentioned.

We are sorry for the mistake. The Ion leakage data were included in previous visions of the manuscript. We removed the data but forgot to remove the corresponding method in the present version.

Materials and methods:To enhance the reproducibility of the study, the authors should provide a more detailed description of the Materials and methods, especially for critical experiments like the Yeast-two-hybrid assays. Clear documentation of specific reagents, strains, and protocols used, along with information on controls, will bolster the validity of the results and facilitate future research in this area.

Thank you for your suggestions. We provided more details in the methods. For yeast two-hybrid assays, the vector information was included in “Vector constructions” section.

Minor Point:Line 61: There is a space between ')' and '.', which needs to be edited.

The space was deleted.

Reviewer #1 (Significance (Required)):This study holds significant importance within the field of plant immunity research. The authors have made valuable contributions through their comprehensive analysis, encompassing genetics, transcriptional, translational, and proteomic approaches, to elucidate the critical role of tRNA thiolation in plant immunity. One of the major strengths of this study lies in its ability to shed light on a previously unknown regulatory mechanism for plant defense. By identifying the cbp mutant and investigating the role of ROL5 and CTU2 in catalyzing the mcm5s2U modification, the authors have unveiled a novel aspect of plant immune regulation. This innovative discovery provides a deeper understanding of the intricate molecular processes governing immunity in plants.Moreover, the study's findings are not limited to the immediate field of plant immunity but also have broader implications for the scientific community. By employing diverse methodologies, the authors have demonstrated how tRNA thiolation exerts control over both transcriptional and translational reprogramming, revealing intricate links between these processes. This integrative approach sets a precedent for future research in the field of plant molecular biology and opens up new avenues for investigating other aspects of immune regulation.In terms of its relevance, the study's findings have the potential to captivate researchers across various disciplines, such as plant biology, molecular genetics, and translational research. The insights gained from this study may inspire researchers to explore further the role of tRNA in other regulation.

Thank you very much for your positive comments and support!

Reviewer #2 (Evidence, reproducibility and clarity (Required)):The authors presented an intriguing and previously unknown mechanism that the tRNA mcm5s2U modification regulates plant immunity through the SA signaling pathway, specifically by controlling NPR1 translation. The manuscript was well-written and logically structured, allowing for a clear understanding of the research. The authors provided strong and persuasive data to support their key claims. However, further improvement is required to strengthen the conclusion that mcm5s2U regulates plant immunity by controlling NPR1 translation.

Thank you very much for your positive comments and support!

Major comments:1. NPR1 translation should be examined to verify the Mass Spec (Figure 5B) and polysome profiling data (Figure 5D) by checking the NPR1 protein and mRNA level using antibodies and qPCR, respectively, in the cgb mutant background to establish a concrete confirmation of CGB regulation in NPR1 translation.

This is a very constructive suggestion. We performed these experiments and found that the transcription levels of *NPR1* were similar between *COM* and *cgb* both before and after *Psm* ES4326 infection (Figure S2), which is consistent with RNA-Seq data. Consistent with the Mass Spec and polysome profiling data, the NPR1 protein level was much higher in *COM* than that in *cgb* (Figure 5C) after *Psm* ES4326 infection. Together, these data further supported our conclusion that translation of NPR1 is impaired in *cgb*.

2. Analyzing the genetic epistasis of CGB and NPR1 to check if CGB regulates plant immunity through the NPR1-dependent SA signal pathway. If the authors' claim is valid, I would expect no addictive effect on bacterial growth in the cgb/npr1 double mutant compared to the single mutants. Due to the broad impact of CGB on plant signaling (Figures 4E and 4F), the SA protection assay, which concentrates on the SA signal pathway, needs to be tested in WT, cgb and npr1 plants as an alternative assay to the genetic epistasis analysis. I expect that the SA-mediated protection is also compromised in cgb mutant background.

Thank you for your suggestions. We did examine the growth of *Psm* ES4326 in the *cgb npr1* double mutant and found that *cgb npr1* was significantly more susceptible than *npr1* and *cgb* (Figure below). Although the additive effects were observed, this result was not against our conclusion for the following reasons. First, the translation of NPR1 was reduced rather than completely blocked in *cgb*. In other words, NPR1 still has some function in *cgb*. But in the *cgb npr1* double mutant, the function of NPR1 is completely abolished, which explains why *cgb npr1* was more susceptible than *cgb*. Second, in addition to NPR1, some other immune regulators (such as PAD4, EDS5, and SAG101) were also compromised in *cgb* (Figure 5B), which explained why *cgb npr1* was more susceptible than *npr1*. Since the result of the genetic analysis was not intuitive, we decided not to include it in the manuscript.

SA signaling is known to regulate both basal resistance and systemic acquired resistance (SA-mediated protection). We have shown that *cgb* is defective in the defect of basal resistance, which *cgb* is sufficient to support our conclusion that the tRNA thiolation is essential for plant immunity. We agree that it is expected that the SA-mediated protection is also compromised in *cgb*. We will test this in the future study.

**Author response image 1. sa2fig1:** 

1. Could the authors comment on why using COM instead of WT as a control to perform the majority of the experiments?

Thank you for your comments. In addition to *ROL5*, the *cgb* mutant may have other mutations compared with WT.*COM* is a complementation line in the *cgb* background. Therefore, the genetic background between *COM* and *cgb* may be more similar than that of WT and *cgb*.

1. In Figure 5E, why does ACTIN2 have an enhanced translation while NPR1 shows a compromised one in cgb mutant? How does the mcm5s2U distinguish NPR1 and ACTIN2 codons? Does mcm5s2U modification have both positive and negative roles in regulating protein translation?

Thank you for raising this question. As previously reported, loss of the mcm^5^s^2^U modification causes ribosome pausing at AAA and CAA codons. Therefore, the translation of the mRNAs with more GAA/CAA/AAA codons (called s^2^ codon) is likely to be affected more dramatically in *cgb*. We have analyzed the percentage of s^2^ codon at whole-genome level (Figure below). The average percentage is 8.5%, while NPR1 contains 10.1% s^2^ codon and actin contains only 4.5% s^2^ codon. When fewer ribosomes are used for translation of the mRNAs with high s^2^ codon percentage, more ribosomes are available for translation of the mRNAs with low s^2^ codon percentage, which may account for the enhanced translation efficiency. To focus on NPR1 and to avoid confusion, we removed the ACTIN data in the revised manuscript.

3. Specify the protein amount used for the in vitro pull-down assay and agrobacteria concentration used for the tobacco Co-IP assay in the protocol section.

We added this information in Method section in the revised manuscript.

4. Delete the SA quantification and Ion leakage assay in the protocol, which are not used in the study.

We are sorry for the mistake. The SA quantification and ion leakage data were included in previous visions of the manuscript. We removed the data but forgot to remove the corresponding method in the present version. We deleted them in the revised manuscript.

5. The strain Pst DC3000 avrRPT2 was not used in this study. Please remove it.

We are sorry for the mistake. The strain Pst DC3000 avrRPT2 was used for ion leakage assay in previous visions of the manuscript. We deleted it in the revised manuscript.

6. In Figure 5F, did the 59 genes tested overlap with the 366 attenuated proteins in the cgb mutant? Were the 59 genes translationally regulated?

Thank you for your suggestion. Venn diagram analysis revealed that 12 genes (about 20%) are also attenuated proteins, suggesting that the mcm^5^s^2^U modification regulates the translation of some SA-responsive genes.

**Author response image 3. sa2fig3:** 

Reviewer #2 (Significance (Required)):The authors' study is significant as it establishes the first connection between tRNA mcm5s2U modification and plant immunity, specifically by regulating NPR1 protein translation. This research expands our understanding of the biological role of tRNA mcm5s2U modification and highlights the importance of translational control in plant immunity. It is likely to captivate scientists working in this field.

Thank you very much for your positive comments and support!

Reviewer #3 (Evidence, reproducibility and clarity (Required)):In this manuscript, the authors identified a cgb mutant that carries a mutation in the ROL5 gene Both the cgb mutant and the newly created rol5-c mutant are susceptible to the bacterial pathogen Psm. The authors showed that ROL5 interacts with CTU2, the Arabidopsis homologous protein of the yeast tRNA thiolation enzyme NCS2. A ctu2-1 mutant is also susceptible to Psm, suggesting the tRNA thiolation may play a role in plant immunity. Indeed, tRNA mcm5S2U levels are undetectable in rol5-c and ctu2-1 mutants. The authors found that the cgb mutation significantly attenuated basal and Psm-induced transcriptome and proteome changes. Furthermore, it was found that the translation efficiency of a group of SA signaling-related proteins including NPR1 is compromised.The manuscript provides solid evidence for the involvement of ROL5 and CTU2 in plant immunity using the rol5 and ctu2 mutants. The authors may consider the following suggestions and comments to improve the manuscript.

Thank you very much for your support and suggestions!

1. The function of the Elongator complex in tRNA modification/thiolation has been extensively studied. In Arabidopsis Elongator mutants, mcm5S2U levels are very low, similar to the levels in the rol5 and ctu2 mutants (Mehlgarten et al., 2010, Mol Microbiology, 76, 1082-1094; Leitner et al., 2015 Cell Rep). In elp mutants, the PIN protein levels are reduced without reduced mRNA levels (Leitner et al., 2015), indicating that Elongator-mediated tRNA modification is involved in translation regulation. The Elongator complex plays an important role in plant immunity, though the reduced mcm5S2U levels in elp mutants were not proposed as the exclusive cause of the immune phenotypes. In fact, it would be difficult to establish a cause-effect relationship between tRNA modification and immunity. These results should be discussed in the manuscript.

Thank you very much for your insightful comment on the role of the ELP complex in tRNA modification and plant immunity. We added a paragraph discussing the ELP complex in the revised manuscript (Line 280-295).

In addition to tRNA modification, the ELP complex has several other distinct activities including histone acetylation, α-tubulin acetylation, and DNA demethylation. Therefore, it is difficult to dissect which activity of the ELP complex contributes to plant immunity. However, the only known activity of ROL5 and CTU2 is to catalyze tRNA thiolation. Considering that the *elp*, *rol5*, and *ctu2* mutants are all defective in tRNA thiolation, it is likely the tRNA modification activity of the ELP complex underlies its function in plant immunity.

2. The interaction between CTU2 and ROL5 in Y2H has previously been reported (Philipp et al., 2014). The same report also showed reduced tRNA thiolation in the ctu2-2 mutant using polyacrylamide gel. These results should be mentioned/discussed in the manuscript.

Thank you for pointing them out. We added this information in the revised version (Line 146-147).

3. tRNA modification unlikely plays a unique role in plant immunity. It can be inferred that mutations affecting tRNA modification (rol5, ctu2, elp, etc.) would delay both internal and external stimulus-induced signaling including immune signaling.

We agree with you that tRNA modification has other roles in addition to plant immunity. In the Discussion section, we have mentioned that “it was found that tRNA thiolation is required for heat stress tolerance (Xu et al., 2020). ……It will also be interesting to test whether tRNA thiolation is required for responses to other stresses such as drought, salinity, and cold.” (Line276-279).

4. It would be interesting to conduct statistical analyses on the genetic codons used in the CDSs whose translation was attenuated as described in the manuscript. Do these genes including NPR1 use more than average levels of AAA, CAA, and GAA codons? If not, why their translation is impaired?

Thank you for your suggestion. We called GAA/CAA/AAA codons s^2^ codon. We have analyzed the percentage of s^2^ codon at whole-genome level (Figure below). NPR1 does contain more s^2^ codon (10.1%) than the average level (8.5%). We are preparing another manuscript, which will report the relationship between s^2^ codon and translation.

**Author response image 4. sa2fig4:** 

Referees cross-commenting

It is important to put current research in the context of available knowledge in the field. The digram in Figure 3C shows that the Elongator complex functions upstream of ROL5 & CTU2 in modifying tRNA. The function of Elongator in plant immunity has been well established. The similarities and differences should be discussed. Additionally, it may no be a good idea to claim that the results are novel.

Thank you for your comments. We added a paragraph discussing the ELP complex in the revised manuscript (Line 280-295). The ELP complex catalyzes the cm^5^U modification, which is the precursor of mcm^5^s^2^U catalyzed by ROL5 and CTU2. In addition to tRNA modification, the ELP complex has several other distinct activities including histone acetylation, α-tubulin acetylation, and DNA demethylation. Therefore, it is difficult to dissect which activity of the ELP complex contributes to plant immunity. However, the only known activity of ROL5 and CTU2 is to catalyze tRNA thiolation. Considering that the *elp*, *rol5*, and *ctu2* mutants are all defective in tRNA thiolation, it is likely the tRNA modification activity of the ELP complex underlies its function in plant immunity. Therefore, our study improved our understanding of the ELP complex in plant immunity. We have deleted the words “new” and “novel” throughout the manuscript.

Reviewer #3 (Significance (Required)):The manuscript provides solid evidence for the involvement of ROL5 and CTU2 in plant immunity. However, the authors did not acknowledge the existing results about the Elongator complex that functions in the same pathway in modifying tRNA. The involvement of Elongator in plant immunity has been well established. The cause-effect relationship between tRNA modification and plant immunity is difficult to demonstrate.

We think that the cause-effect relationship between the activities of the ELP complex and plant immunity is difficult to demonstrate because the ELP complex has several distinct activities other than tRNA modification. However, since the only known activity of ROL5 and CTU2 is to catalyze tRNA thiolation, the cause-effect relationship between tRNA thiolation and plant immunity is clear, which indicated that the tRNA modification activity of the ELP complex contributes to plant immunity.